# A Teledentistry Pilot Study on Patient-Initiated Care

**DOI:** 10.3390/ijerph19159403

**Published:** 2022-07-31

**Authors:** Clare Lin, Nuno Goncalves, Ben Scully, Ruth Heredia, Shalika Hegde

**Affiliations:** Dental Health Services Victoria, The Royal Dental Hospital of Melbourne, 720 Swanston Street, Carlton, VIC 3053, Australia; clare.lin@dhsv.org.au (C.L.); nuno.goncalves@dhsv.org.au (N.G.); ben.scully@dhsv.org.au (B.S.); ruth.heredia@dhsv.org.au (R.H.)

**Keywords:** Teledentistry, Telehealth, dentistry, public health, workforce, COVID-19

## Abstract

COVID-19 has challenged the public dental workforce in their ability to continue providing routine oral health care services. To mitigate the risk of COVID-19 transmission to staff and patients, Teledentistry was implemented in many parts of the world, mainly to provide remote consultations, undertake triage, and offer preventive educational sessions. The aim of this paper is to describe Dental Health Services Victoria’s (DHSV) patient-initiated Teledentistry model of care implemented during peak COVID transmission in Victoria. The Teledentistry model supported patient-centered care involving active collaboration and shared decision making between patients, families, and clinicians in designing and managing remote care plans. DHSV’s eligible patient cohort includes disadvantaged population groups with greater oral health needs. Strong emphasis was placed on the simplicity and user friendliness of the Telehealth platform, as well as the support for patients with low technology literacy. Consumers and dental workforce were consulted and modifications to the use of language and services were undertaken before the launch. A total of 2492 patients accessed Telehealth services between May 2020 and April 2021. Approximately 39% of patients were born in a country other than Australia. A total of 489 patient-reported experience measures (PREMs) were received. Patients agreed or strongly agreed that the care they received met their needs (87%); they received answers to their questions (89%); they left their visit knowing what is next (87%); they felt they were taken care of during their visit (90%); and they felt involved in their treatment (89%). Teledentistry enabled patients to initiate access to care and consult with dental workforce remotely and safely during peak pandemic.

## 1. Introduction

Healthcare innovations in the field of information and communications technology (ICT) have seen growing interest in the use and adoption of Telehealth. Teledentistry is a branch of Telehealth that is specifically dedicated to dentistry and uses ICT, electronic health records, digital imaging, and the Internet for consultation, supervision, monitoring, or continuing of education [1]. The use of Teledentistry for screening oral diseases, determining treatment needs, and providing timely access to oral health care workforce is promising [1,2].

Innovative oral health workforce models have been effectively tested using Teledentistry, specifically to undertake distant triaging, diagnosing, patient referrals, and offering preventive care [3,4]. Several studies conducted in a range of countries have reported on the positive workforce and patient perceptions on the effectiveness and applications of Teledentistry [5,6,7]. These included remote access to dental workforce, remote triaging and treatment planning, a reduction in travel time and the number of face-to-face appointments, an improvement in clinical workflows, and a reduction in non-essential interactions during the pandemic. Evidence shows that, across the globe, Teledentistry has been successfully adopted by the oral healthcare workforce, with effective diagnostic performance in identifying dental caries in children [8,9], diagnosing oral lesions [10], screening for oral trauma [11], performing orthodontic consultations [12], and undertaking referrals [13]. A number of quality studies, including studies with control groups, reported similar or better clinical outcomes using Teledentistry when compared to conventional face-to-face interventions [2].

There is a consistent trend in the literature supporting the efficacy and effectiveness of Teledentistry [9,10,14]. This has led to the steady implementation and use of Teledentistry, particularly in the USA, Australia, India, and Brazil, as well as several countries in Europe [8,11,12,13,15,16,17]. Systematic reviews show that although there is heterogeneity between studies from different countries in terms of study designs, clients, workforce, settings, and outcomes, a trend exists supporting the efficacy and effectiveness of Teledentistry [2].

Dental Health Services Victoria (DHSV) is the lead public oral health agency in Victoria that provides oral healthcare to eligible population through the Royal Dental Hospital of Melbourne (RDHM) by funding 53 community dental agencies across the state to deliver oral healthcare locally. As COVID-19 restrictions were introduced across Victoria in March 2020, limitations on dental services were implemented to minimize the public health risks of COVID-19. In alignment with the government restrictions, DHSV suspended all non-essential and routine aerosol-generating dental procedures to reduce the risk of COVID-19 transmission. As per the Australian Health Protection Principal Committee recommendations, dental practices operated within a framework of practice restrictions and guidance produced by the Australian Dental Association (ADA Inc., Toronto, ON, Canada). The essential and routine dental procedures were constantly changing and adapting to the levels of COVID-imposed restrictions. During the peak of the pandemic, dental services operated on a much higher restriction level than private dental services. The COVID-19-initiated dental restrictions presented two immediate concerns for DHSV. Firstly, patients who needed urgent care may not be identified and miss timely care for emergency conditions. Secondly, patients who did not need urgent care might present to public dental clinics, exposing themselves and the workforce to the virus.

The aim of this paper is to describe DHSV’s patient-initiated Teledentistry model of care implemented during peak COVID transmission in Victoria. Against the backdrop of COVID-19, this article describes the origin, rationale, and scope of DHSV’s Teledentistry service to continue to deliver public oral healthcare. The article presents the process, benefits, limitations, and learnings of establishing the Teledentistry service at DHSV. 

## 2. Materials and Methods

### 2.1. Implementation of Teledentistry at DHSV

In April 2020, DHSV established a Telehealth working group to plan the implementation. The working group comprised eleven members representing a range of areas (ICT (*n =* 2), clinical (*n =* 3), clinical system analysts (*n =* 2), administration (*n =* 2), and patient liaison services (*n =* 2)). The Teledentistry launch was planned to complement the ongoing emergency and urgent face-to-face care that the public dental sector continued to provide throughout the pandemic under strict infection control and prevention protocols. The working group was tasked with reviewing the Teledentistry requirements holistically to transition to a platform that would be able to deliver the below three stream scenarios.

*Stream 1—Patient-initiated care*: Support patient-initiated care by providing RDHM with the ability to remotely triage and manage patients.

*Stream 2—Clinician-to-clinician-initiated care:* Enable clinician-to-clinician video consultations related to patient care.

*Stream 3—Clinician-to-service-initiated care:* Enable remote appointments and emergency follow-up reviews via video consults and active support for the community to manage and maintain good oral health.

The Telehealth working group mapped the patient journey, technological capabilities, administrative tasks, workforce requirements, and clinical processes needed to meet the objectives of the three streams. With the initial need identified as greatest for Stream 1 (patient-initiated care), resources and attention were first focused there. This paper will describe DHSV’s *Stream 1 patient-initiated care* and the feasibility of implementing teleconsultation and telediagnosis in a public dental system.

### 2.2. Telehealth Platform

In consultation with the Victorian Department of Health, DHSV opted to adopt the Healthdirect platform which integrated strong security and privacy compliance requirements. DHSV’s eligible patient cohort includes vulnerable populations with greater oral health needs who are at higher risk of poor oral health. These population include people on low income, Aboriginal and Torres Strait Islander people, people who are homeless or at risk of homelessness, pregnant women, refugees and asylum seekers, and people registered with mental health and disability services [18]. Due to the diverse make-up of DHSV’s patient cohort, strong emphasis was placed on the simplicity and user friendliness of the Teledentistry platform, as well as support for patients with low technology literacy. The Healthdirect platform was able to meet the organizational requirements. The ICT team established the Healthdirect platform at DHSV and ran training and trial sessions of the mapped Teledentistry processes. In parallel to the establishment of the Teledentistry service, DHSV’s consumer engagement and co-design consultant worked with the consumer advisory network to understand consumers’ views on Teledentistry. Based on consumer feedback, subsequent modifications to the use of language and services were undertaken.

### 2.3. Medico-Legal Implications

In accordance with the Victorian legislation and Government policy, DHSV upholds and protects the privacy and confidentiality of patient information it holds, and uses it only for the intended purpose of providing patient care. Health-grade privacy, security, and data protection are fundamental to the Healthdirect platform, including its video enabled call model. The Healthdirect video calls for compliance with the Australian government privacy policies. By default, the video calls do not retain any identifiable patient information and patients do not leave a digital footprint on the platform. All data, including live video calls, are encrypted.

Clinicians are trained and responsible for protecting their patient’s privacy and their rights to confidentially. Patient consent for consultation, including the acknowledgement of Teledentistry limitations (e.g., the inherent risk of improper or limited diagnosis and/or treatment) is attained and recorded. Clinicians use their judgement with regards to the safety and clinical appropriateness of the technology-based patient consultations and decide whether a direct face-to-face examination is necessary. Patient identity is confirmed and recorded using at least three patient identifiers, such as name, age/date of birth, and address before proceeding. Video services are used to view client’s identifying documents and concession card details when registering new clients, updating details, and confirming identity.

### 2.4. Launch of Teledentistry Patient Service Delivery Model

A soft launch of the Teledentistry service took place in May 2020, with one administrative officer and a clinician allocated to the Monday to Friday service from 9.00 a.m. to 5.00 p.m. A live link was placed on the DHSV website which patients could click on to connect with ‘on demand’ services via the Healthdirect platform. The launch was intentionally gradual, allowing troubleshooting and processes to be refined among a small number of early users.

Figure 1 describes the patient service delivery model for Teledentistry services. There are two potential entry points to the service: a patient may have a Teledentistry appointment booked via the call centre or the walk-in clinic at RDHM, or the patient may initiate the interaction on-demand by clicking on the Teledentistry link on the DHSV website. The patient is then taken to a standard screen where they will have to enter their personal details (name, phone number, and concession card details). Once the patients read, understand, and acknowledge the privacy policy and the Australian Charter of Healthcare Rights, they are taken to the Healthdirect waiting room. An administrative officer commences the video call, speaks with the patients, and performs required administrative tasks before transferring the patient to the clinician. The clinician obtains informed consent for the Teledentistry consult and performs a rapid clinical assessment and triage. The rapid clinical assessment involves documenting the patient’s history, their medical history, and presenting complaint. A limited video examination may be conducted depending on the quality of the connection. Based on this assessment, the clinician determines the patient’s triage category using a standardized rubric which the clinicians are trained to use to ensure consistent triaging. Data collected across the public dental sector in Victoria are standardized. This includes standard definitions, item codes, and data inputs into the Titanium patient management system.

Depending on the triage category, and the level of dental restrictions in place due to COVID-19, the clinician may recommend the following to the patient: attend face-to-face care at RDHM or a community dental agency; provide self-management advice (e.g., prescriptions, oral health instruction, smoking cessation advice or reassurance); arrange referrals (e.g., for OPG X-ray, to specialist services or social work); or place the client on an appropriate waitlist. In-person appointments are offered to patients depending on the level of restrictions dental services are allowed to operate. Clients who are advised to attend public dental services for urgent treatment are required to complete a COVID-19 screening questionnaire conducted via Telehealth. To capture patient experience, all patients who access Teledentistry services are sent a patient-reported experience measures (PREMs) survey following their appointment.

## 3. Results

### 3.1. Teledentistry Patient Service Utilisation

A total of 2492 patients accessed Teledentistry services at RDHM between 1 May 2020 and 30 April 2021. A higher proportion of patients aged between 25 and 44 years (*n =* 998, 40%), followed by patients aged between 45 and 64 years (*n =* 596, 24%), accessed the services, accounting for almost two-thirds of the appointments. Approximately 5% (*n =* 115) of the patients identified themselves as Aboriginal and/or Torres Strait Islanders and 3% (*n =* 69) as refugees and asylum seekers. About 39% (*n =* 962) of patients were born in a country other than Australia. Most patients (93%, *n =* 2326) resided in metropolitan areas, with 65% (*n =* 1624) residing in either Northern or Western metropolitan area (Figure 2 and Figure 3).

### 3.2. Teledentistry Attendance Rates

Overall, 84% of patients who initiated Teledentistry services attended their appointments. Attendance was high (94.1%, *n =* 963) if the patient was previously booked for a non-Teledentistry appointment. A 5.9% (*n* = 60) failure-to-attend rate was observed. Patient cohorts who showed higher attendance rates (≥85%) included children aged 12 years and below; people who did not hold a concession card; and people who resided in Eastern metro, Southern metro, Grampians, Loddon, and Gippsland areas. (Figure 4 and Table 1).

### 3.3. Telehealth Patient-Reported Experience Measures

Between July 2020 and April 2021, a total of 489 PREMs responses were received from patients who accessed Teledentistry services. Patients agreed or strongly agreed that the care they received met their needs (87%), they received answers to their questions (89%), they left their visit knowing what is next (87%), they felt they were taken care of during their visit (90%), and they felt involved in their treatment (89%). Approximately 56% of the patients felt that the quality of treatment they received was excellent and 28% felt it was good. Nearly 83% of patients indicated they would recommend DHSV’s Teledentistry service to family or friends.

In addition, patients provided feedback on the service using the free text functions of PREMs. The feedback received were largely positive. Key themes included the ease, convenience, and time-saving nature of the Teledentistry service, as well as the professionalism and kindness of staff. Patients particularly appreciated the contact with clinicians, and the support and reassurance they received during COVID-19 restrictions. Active collaboration and shared decision making between patients, families, and clinicians in designing and managing remote care plans was perceived as a success by both the patients and the clinicians. While criticisms were less frequent, common barriers identified by patients included internet connectivity issues, restricted access to face-to-face consultation under COVID-19 restrictions, and long waiting times before being able to receive general and specialist care.

## 4. Discussion

### 4.1. Reducing Oral Health Inequities through DHSV’s Teledentistry Services

A substantial body of literature shows several examples of successful implementation of Teledentistry services within public dental systems during COVID-19 [13,19]. Teledentistry has been identified as one of the most viable tools to address oral health inequities by increasing access to care for vulnerable and underserved populations, reducing barriers to accessing dental workforce, improving oral health outcomes, and increasing the use of oral healthcare [15,17,20]. Inequities in the provision of oral healthcare and difficulties accessing oral health services are major public health challenges for a large proportion of people from disadvantaged backgrounds [15,20]. The utility of Teledentistry in potentially addressing the service delivery gap, reducing oral health inequalities, and providing sustainable workforce solutions is highlighted in several studies [1,14,17,21]. Evidence shows that despite initial start-up costs, Teledentistry can assist in reducing inequalities in oral health [22]. 

Evaluation studies on the use of Teledentistry from a patient’s perspective showed that Teledentistry helped patients to seek access to healthcare earlier, provide access to specialist care, minimize time off work, and reduce travel over long distances to receive consultations [23]. From a workforce perspective, Teledentistry has the potential to eliminate inappropriate referrals [24], reduce long waiting lists for specialist consultations [20], and provide screening and referrals for vulnerable and underserved populations [14]. Evidence shows improved clinical outcomes following Teledentistry intervention and high workforce satisfaction with Teledentistry in a range of settings [2].

### 4.2. Challenges and Lessons from Implementing Teledentistry

While many countries have adopted Teledentistry in the face of a surge in cases of COVID-19, the overall adoption of Teledentistry by the workforce has been slow and inconsistent across the world [2]. The reasons for this are driven by many factors such as professional readiness, low technology literacy, a lack of clear intraoral imaging, IT connectivity and outages, barriers for patients with low technology literacy, difficulties providing prescriptions remotely to pharmacists for class X drugs, and a lack of financial compensation [5,7,15,17]. Additionally, the effectiveness of Teledentistry services depends on extensive workforce planning, the development of user-friendly technical processes and systems, and targeted training for the workforce prior to implementation [13,17,25]. A study assessing the impact of Teledentistry, as well as its application and trends in uplifting dental practice and clinical care around the world, found that about 50–70% of dental professionals expressed their concerns regarding the security of the data and consent of patients; however, they endorsed Teledentistry as a useful tool for improving clinical practice as well as patient care [6].

An obvious challenge documented with our Teledentistry model is the limited ability to perform oral health examinations and the inability to palpate and undertake diagnostic tests. This meant that definitive diagnoses could not always be reached. Additionally, patients who used phones to connect to the Telehealth platform had poor image resolution and connectivity performance. While video consultations are not as accurate as conventional oral health examinations, they can be sufficient for providing a rapid and relevant diagnosis [20]. 

Clinicians found that communicating through Telehealth platform felt more challenging than in-person care. However, the visual component of Healthdirect provided advantages over a purely audio consultation though phones. There were additional challenges noted for patients with language barriers and patients with hearing and speech impairment. In these situations, the chat function with the Healthdirect platform was useful in supporting client consultations. In our pilot the response from workforce on the use of Teledentistry has been largely positive, with most users becoming increasing comfortable with the platform and technology over time. 

### 4.3. Sustainability of Teledentistry

The successful implementation of a sustainable Teledentistry model is a complex and collaborative process, involving numerous factors at individual, infrastructure, and organizational levels. Costing and economic analysis for DHSV’s Teledentistry initiative have not yet been completed. This may impact the financial viability of the service. However, DHSV’s experience does point to potential cost-savings through the reduced overhead costs of some services which could be provided via Teledentistry rather than onsite, as well as increased capacity to care for patients without a corresponding increase in dental chairs or physical infrastructure.

Studies examining different Teledentistry applications have found that technology can be successfully integrated into different workforce settings; however, there is a lack of good-quality evidence on its application on a workforce level [2]. The existing workforce policy and practice of Teledentistry lack support among policy-makers due to the lack of evidence related to the cost-effectiveness of Teledentistry in head-to-head comparisons [17]. A study in Australia showed that the Teledentistry model of dental screening can minimize costs by comparing the cost of in-person and remote Telehealth dental screening of school children by the mid-level workforce, such as dental therapists. [19] The estimated staff salary saved with the Teledentistry model was AUD 56 million, and the estimated travel allowance and supply expenses avoided were AUD 16 million and AUD 14 million, respectively, representing an annual reduction of AUD 85 million in total. Expanding the roles of mid-level workforce and allowing the workforce to work across their top scope of practice can increase the number of oral healthcare providers performing screenings, delivering care, and offering referrals using Teledentistry [2].

While Telehealth applications are becoming increasingly popular in dentistry, studies on the assessments of clinical outcomes, economic analyses, and sustainability of Teledentistry are limited [13]. Although there is emerging evidence supporting the efficacy of Teledentistry, there is a lack of conclusive evidence, particularly for its cost-effectiveness, long-term use, and sustainability, to inform evidence-based policy decisions on Teledentistry [14]. Regardless of the lack of good-quality evidence supporting the cost-effectiveness of Teledentistry, existing evidence indicates that Teledentistry, even with additional costs, helps in reducing inequalities in the provision of health care [11]. While Teledentistry is a cost-effective modality for both patients and providers, finding sustainable funding to provide the services is often challenging.

## 5. Conclusions

Teledentistry offers a useful platform to provide patient-centered care. Further studies are warranted to assess the effectiveness of Teledentistry programs in measuring health outcomes that matters to patients.

## Figures and Tables

**Figure 1 ijerph-19-09403-f001:**
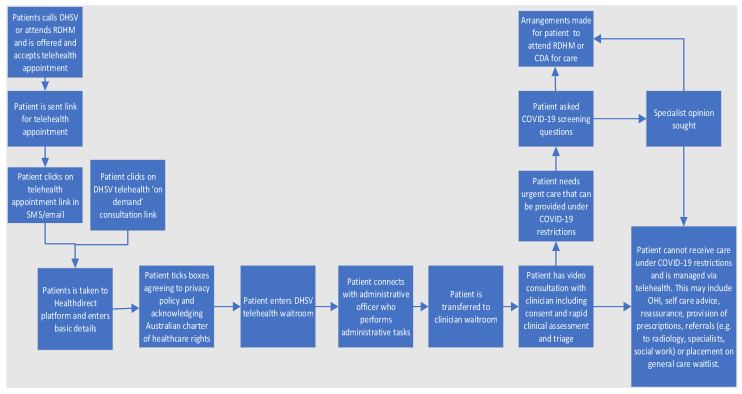
DHSV’s patient-initiated Teledentistry model of care.

**Figure 2 ijerph-19-09403-f002:**
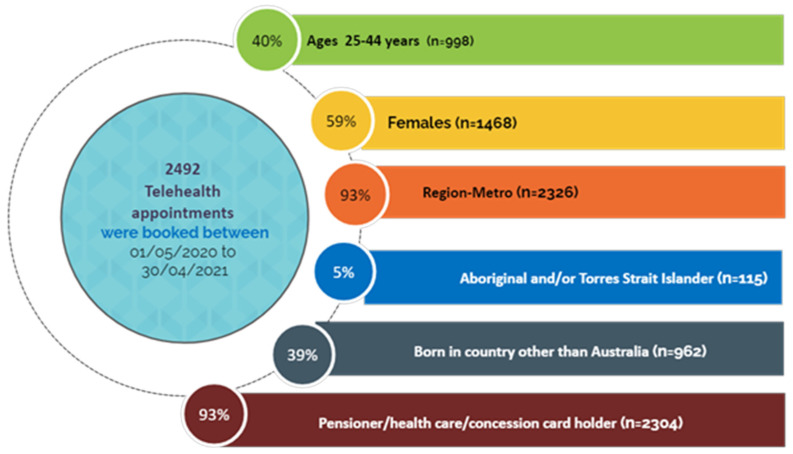
The number of Teledentistry appointments booked at RDHM between 1 May 2020 and 30 April 2021 (*n =* 2492).

**Figure 3 ijerph-19-09403-f003:**
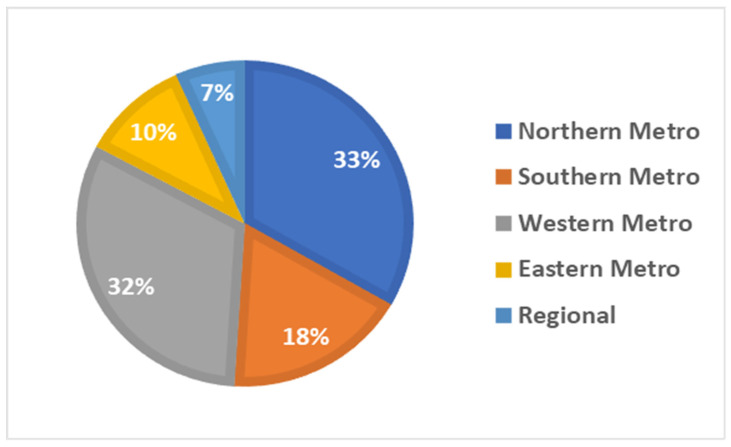
The percentage of Teledentistry appointments by Victorian region (*n =* 2492).

**Figure 4 ijerph-19-09403-f004:**
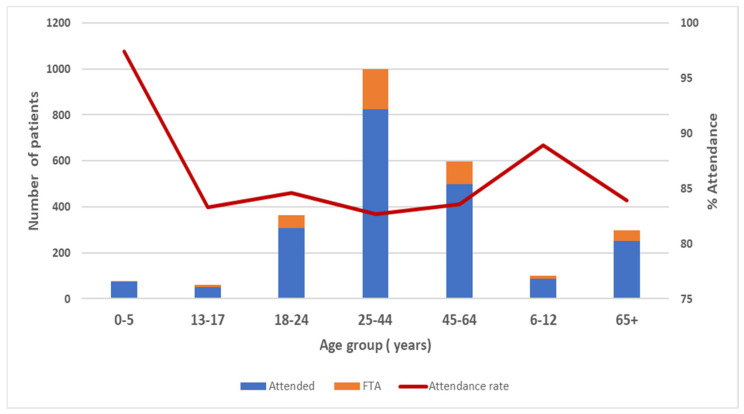
Teledentistry attendance and failure-to-attend (FTA) rates by age group.

**Table 1 ijerph-19-09403-t001:** Client demographics for RDHM telehealth appointments (attended/FTA) between 1 May 2020 and 30 April 2021.

Variable Category	Variable Category	Attended Appointment *n* (%)	FTA Appointment *n*(%)	All * *n* (%)	% Attendance	% FTA
Overall sample		2094	398	2492	84.0	16.0
Age groups (years)	0–5	75 (3.6)	2 (0.5)	77 (3.1)	97.4	2.6
	6–12	88 (4.2)	11 (2.8)	99 (4.0)	88.9	11.1
	13–17	50 (2.4)	10 (2.5)	60 (2.4)	83.3	16.7
	18–24	307 (14.7)	56 (14.1)	363 (14.6)	84.6	15.4
	25–44	825 (39.4)	173 (43.5)	998 (40.1)	82.7	17.3
	45–64	498 (23.8)	98 (24.6)	596 (23.9)	83.6	16.4
	65+	251 (12.0)	48 (12.1)	299 (12)	83.9	16.1
Gender	Male	868 (41.5)	153 (38.4)	1021 (41)	85.0	15.0
	Female	1224 (58.5)	244 ()	1468 (58.9)	83.4	16.6
	Other	2 (0.1)	1 (0.3)	3 (0.1)	^#^ 66.7	^#^ 33.3
Card status	Healthcare card	1029 (49.1)	194 (48.7)	1223 (49.1)	84.1	15.9
	Pensioner card	895 (42.7)	182 (45.7)	1077 (43.2)	83.1	16.9
	DVA pens card	1 (0.1)	3 (0.75)	4 (0.2)	^#^ 25	^#^ 75
	No card	169 (8.1)	19 (4.8)	188 (7.5)	89.9	10.1
Country of birth	Australia	1225 (58.5)	198 (49.8)	1423 (57.1)	86.1	13.9
	Not Australia	799 (38.2)	163 (40.9)	962 (38.6)	83.1	16.9
	Not stated or Inadequately described	70 (3.3)	37 (9.3)	107 (4.3)	65.4	34.6
Preferred language	English	1893 (90.4)	361 (90.7)	2254 (90.5)	84.0	16.0
	Not English	185 (8.8)	34 (8.5)	219 (8.8)	84.5	15.5
	Not stated/ Inadequately described	16 (0.8)	3 (0.8)	19 (0.8)	84.2	15.8
Interpreter required	Yes	128 (6.1)	28 (7.0)	156 (6.3)	82.1	17.9
	No	1966 (93.9)	370 (93)	2366 (93.7)	84.2	15.8
Indigenous status	ATSI	93 (4.4)	22 (5.5)	115 (4.6)	80.9	19.1
	Non-ATSI	1954 (93.3)	374 (94)	2328 (93.4)	83.9	16.1
	Not stated	47 (2.2)	2 (0.5)	49 (2)	95.9	4.1
Refugee/ Asylum seeker	Yes	57 (2.7)	12 (3.0)	69 (2.8)	82.6	17.4
	No	2037 (97.3)	386 (97)	2423 (97.2)	84.1	15.9
Eligible child/ young person	Yes	194 (9.3)	19 (4.8)	213 (8.6)	91.1	8.9
	No	1900 (90.7)	379 (95.2)	2279 (91.5)	83.4	16.6
Eligible pregnant woman	Yes	13 (0.6)	3 (0.8)	16 (0.6)	81.3	18.8
	No	2081 (99.4)	395 (99.3)	2476 (99.4)	84.1	15.9
Residence	Private res.	2051 (98)	387 (97.2)	2438 (97.8)	84.1	15.9
	Homeless	30 (1.4)	10 (2.5)	40 (1.6)	75	25
	Supp. res. care	5 (0.2)	1 (0.3)	6 (0.2)	83.3	16.7
	Res. aged care	3 (0.1)	0	3 (0.1)	^#^ 100	0
	^!^ Other accom	1 (0.1)	0	1 (0.04)	^#^ 100	0
	^ Not stat/Inad	4 (0.2)	0	4 (0.2)	^#^ 100	0
Residential region	Northern Metro	696 (33.2)	134 (33.7)	830 (33.3)	83.9	16.1
	Southern Metro	378 (18.1)	63 (15.8)	441 (17.7)	85.7	14.3
	Western Metro	647 (30.9)	147 (36.9)	794 (31.9)	81.5	18.5
	Eastern Metro	228 (10.9)	33 (8.3)	261 (10.5)	87.4	12.6
	Barwon	26 (1.2)	5 (1.3)	31 (1.2)	83.9	16.1
	Gippsland	39 (1.9)	3 (0.8)	42 (1.7)	92.9	7.1
	Grampians	16 (0.8)	1 (0.3)	17 (0.7)	94.1	5.9
	Hume	36 (1.7)	9 (2.3)	45 (1.8)	80	20
	Loddon	23 (1.1)	2 (0.5)	25 (1)	92	8
	^+^ Un/Interstate	5 (0.2)	1 (0.3)	6 (0.2)	83.3	^#^ 16.7

* Rounding may affect % totals; ^#^ low numbers; ^!^ accommodation otherwise not classified; ^ not stated/inadequately described; ^+^ unknown/interstate.

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
