# Peer review of "A Teledentistry Pilot Study on Patient-Initiated Care"

_ijerph, 2022, doi:10.3390/ijerph19159403_

Round 1
Reviewer 1 Report
Dear Authors,
it was with pleasure that I read the manuscript regarding the concept teledentistry services. It fulfils the requirements expected of a concept paper, only there are some points at which I believe you should be more specific.
Lines 38-39 - please specify what you mean by "positive perceptions" - you write a little bit more later on, only it should be more specific.
Lines 69-70 - what were the "essential and routine dental procedures"? It varied across countries, so please - specify that,
Lines 104-105 - please, add references/specify what populations are at higher risk
Lines 155-156 - what did the training regarding standarization look like?
In pages 4 and 5 there was no information regarding the waiting time between the call/tele-appointmend and the in-person appointment, if it was needed - please specify that
I would strongly recommend to either leave out some information from the Tables and present it in a chart form, as there is a lot of text that seems difficult to follow.
In paragraph 3.3, could you please specify/try to systematize how patients with different problems reacted to teledentistry appointment?
Lines 265-271 - The first sentence is trivial. Please, re-write this paragraph in a way that sounds more appropriate for a scientific paper (e.g. the paragraph above is much better)
Lines 301-317 - e.g. line 303 - what reduction? 5%? 10%?
In conclusion, I believe the manuscript is eligible for publication after improvements regarding the level of detail and specificity are implemented.
Kind regards.
Reviewer 2 Report
Type of paper:
Why is this paper a so-called concept paper? What qualifies it as such?
Title:
Title should be shortened; it could also be called as “…a pilot study on patient-initiated care”.
Abstract:
"COVID-19 has challenged the public dental workforce in their ability to continue to pro- 9 vide routine oral health care services. Across the globe teledentistry was seen as a viable option to 10 mitigate the risk of COVID-19 to workforce and patients. " This statement is misleading because it implies that teledentistry could be used to treat patients.
The patient-centered care model should be explained and presented first.
Aim of the study is not clear and should be formulated.
What is meant by telemedicine services that were used?
What is meant by this sentence "Harnessing technological capabilities enabled DHSV to improve patient access to dental 23 workforce during COVID-19." ?
Keywords:
ok
Introduction:
Subdivision with 1.1 is not necessary
Materials and Methods:
The working group should be specified, how many participants, which professions, etc.?
What standardization for data collection or calibration was done?
Results section is missing -> should be redesigned
Figure 2 is not labeled and difficult to read. What is the benefit of Figure 2?
Discussion:
Discussion is too long, should be shortened. The points strengths and limitations as well as the generalization to other regions should be clarified.
Conclusion:
Conclusion should be clearly defined as a separate point.
The article is about one aspect of teledentistry, a very interesting topic. However, the paper needs a deep revision. The points mentioned should be considered. The aim should be clearly defined, and the focus of the study should be specified. Currently, purely descriptive data are presented, but without a results section in the paper. Exploratory questions could also be incorporated, and legal spelling errors corrected. Since the study is thematically linked between Teledentistry and COVID-19, it is recommended that the situation in Australia be supported with concrete data/numbers.
Round 2
Reviewer 2 Report
Ok